# Electrodeposited magnetic nanoporous membrane for high-yield and high-throughput immunocapture of extracellular vesicles and lipoproteins

Chenguang Zhang [1,6], Xiaoye Huo[1,6], Yini Zhu[2], James N. Higginbotham[3], Zheng Cao[3], Xin Lu [2], Jeffrey L. Franklin[3,4], Kasey C. Vickers[3], Robert J. Coffey[3,4], Satyajyoti Senapati[1], Ceming Wang [5✉] & Hsueh-Chia Chang [1✉]

Superparamagnetic nanobeads offer several advantages over microbeads for immunocapture of nanocarriers (extracellular vesicles, lipoproteins, and viruses) in a bioassay: high-yield capture, reduction in incubation time, and higher capture capacity. However, nanobeads are difficult to "pull-down" because their superparamagnetic feature requires high nanoscale magnetic field gradients. Here, an electrodeposited track-etched membrane is shown to produce a unique superparamagnetic nano-edge ring with multiple edges around nanopores. With a uniform external magnetic field, the induced monopole and dipole of this nano edge junction combine to produce a 10× higher nanobead trapping force. A dense nanobead suspension can be filtered through the magnetic nanoporous membrane (MNM) at high throughput with a 99% bead capture rate. The yield of specific nanocarriers in heterogeneous media by nanobeads/MNM exceeds 80%. Reproducibility, low loss, and concentration-independent capture rates are also demonstrated. This MNM material hence expands the application of nanobead immunocapture to physiological samples.

[1] Department of Chemical and Biomolecular Engineering, University of Notre Dame, Notre Dame, IN 46556, USA. [2] Department of Biology, University of Notre Dame, Notre Dame, IN 46556, USA. [3] Department of Medicine, Vanderbilt University Medical Center, Nashville, TN 37232, USA. [4] Department of Cell and Developmental Biology, Vanderbilt University School of Medicine, Nashville, TN 37232, USA. [5] Aopia Biosciences, 31351 Medallion Dr, Hayward, CA 94544, USA. [6] These authors contributed equally: Chenguang Zhang, Xiaoye Huo. ✉email: cwang9@nd.edu; hchang@nd.edu

Extracellular vesicles (EVs) and lipoproteins are biological nanoparticles that can be found in various biological fluids[1–3]. Their recently discovered function of delivering molecular cargo between cells has catalyzed considerable research activity in many fields[4,5]. These biological nanocarriers may be critical mediators of intercellular communication[6,7]. Thus, specific EVs, lipoproteins, and their molecular cargos are also potential disease biomarkers[8–10]. However, these biomarkers are often not unique to diseased cells but are simply overexpressed. Consequently, precise quantification is required. Due to their size and heterogeneity, high-yield isolation of specific EVs and lipoproteins remains challenging and may introduce substantial bias in the biomarker assay[11–15]. EVs and lipoproteins also tend to degrade, aggregate, or be adsorbed in many devices[16,17]. Thus, immediate and short-contact isolation is preferred over flow cytometry and chromatography separation, whose pre-treatment/separation processes are long and complex. An effective, rapid, and accessible isolation method is hence a prerequisite for any clinical application involving EVs and lipoproteins. Advances in high-yield capture technologies are beneficial across many biomedical spaces, including for the detection of pathogenic viruses or bacteria.

The most specific EV and lipoprotein isolation method is immunocapture;[18–20] however, traditional immuno-capture technologies, like immunoprecipitation (IP) and immunoaffinity chromatography, have low-yield issues due to probe saturation and analyte loss. If the nanocarriers are fluorescently labeled, those captured by magnetic microbeads can be sorted and quantified by flow cytometry. However, the labeling and isolation process is time-consuming and may require more than one day to achieve the optimal yield, resulting in significant nanocarrier loss. Their throughputs are also limited by the long incubation time (8–48 h) because of the low mobility of microbeads for the transport-limited docking reaction. A solution to the yield and incubation time issues is to use nanomagnetic beads. Their large surface area per volume provides more binding sites. Their smaller size leads to higher diffusivity and a shorter incubation time (~30 min). The beads can also diffuse through a heterogeneous physiological sample to capture specific nanoparticle targets that have reduced mobility due to complexification or aggregation. Their large surface area per volume provides more binding probes by a factor equal to the ratio of the microbead/nanobead radii (~100) for the same bead weight concentration. This increase in the probe number can lead to the complete depletion of all the target nanocarriers, particularly if the antibody probes have high affinity, thus providing orders of magnitude higher nanocarrier binding yield.

However, due to their superparamagnetic nature, it is difficult to trap nanobeads and their captured nanocarriers after bulk immunocapture. The magnetic force on the superparamagnetic beads is proportional to the gradient of the field squared (twice the product of the field and field gradient), whereas the force on a magnetic microbead is proportional to the local field. For the commonly used magnetic microbead traps, the field gradient is confined to less than one radius of the microbead and hence, can only trap nanobeads within a small area around the microbead. Therefore, a long column of densely packed beads is required for high-yield capture. For example, commercial microbeads columns (μColumn, Milyteni Biotec) for nanobead capture only trap 20–30% of the nanobeads[21]. Repeated (>4 ×) trapping is necessary to produce >90% yield. A magnetic film can produce a higher field penetration length than a magnetic bead due to its non-focusing (non-radial) geometry. Recently, Issadore and colleagues developed a magnetic layer-coated nanoporous membrane with improved capture yield, but multiple layers of membranes are still required for efficient bead capture[22]. Although the field is long-range, the field gradient is not for a planar magnetic film, except at corners. In our earlier work on electric fields at microchannels[23] and nanopores[24], we showed that a singular electric field with a high gradient occurs in the high-permittivity (water) side of a wedge corner of a channel or a pore if the wedge angle $\alpha$ of the higher permittivity phase exceeds $\pi$. This wedge singular field decays radially from the wedge tip with a power-law scaling of $-(\pi/\alpha) - 1$ and hence also has a high-field gradient. The radial decay exponent is bound between $-2$ of a sphere and $-3/2$ of an infinitely long cylinder. This singular wedge mode is antisymmetric around the wedge and introduces a dipole in the high-permittivity phase. There is a more well-known "lightning rod" wedge singularity in near-field plasmonics[25–27] that is symmetric around the wedge, with the singular field occurring on the low-permittivity side. It occurs when the high-permittivity side has a wedge angle that is less than $\pi$. It introduces a monopole on the low-permittivity side of the wedge. Herein, we extend this concept to magnetic fields to achieve high-yield capture of superparamagnetic beads with high throughput. We designed a multi-edge superparamagnetic NiFe nanoedge with a heterogeneous junction, whose edges sustain both a magnetic monopole and dipole around each nanopore of a nanoporous polymer membrane. This approach will significantly increase the capture yield of one membrane to 99% at a throughput of 5 mL/h for a single magnetic nanoporous membrane (MNM).

Compared to the smooth pore edge formed during sputtering, edges on the electroplated membrane are sharper to approach the wedge geometry. Therefore, electroplating was used instead of sputtering for $Ni_{80}Fe_{20}$ layer deposition (Fig. 1a). Moreover, because of the high field at the Au film junction during electroplating, the NiFe film wraps around the gold layer sputtered inside the pore to form the desired edge geometry for a dipole. The uncaptured EVs can go through the straight pores and be collected in the flow-through (Fig. 1b). We proved MNM's efficiency and specificity using high-density lipoproteins (HDL) as a model and observed that >80% of HDL is recovered using the method, nearly doubling the recovery rate for commercial kits. We also demonstrated that MNM has a high and consistent yield and hence, can provide the necessary statistics for quantifying biomarkers carried by EVs and lipoproteins in heterogeneous physiological fluids (Fig. 1c).

## Results

### Theory and simulation of heterogeneous superparamagnetic nano-junction.
Because the magnetic moment of a superparamagnetic nanobead is induced by the external field, the force on it is described by:

$$\vec{F}_M = \frac{\chi_{eff} V \mu}{2\mu_0} \nabla |\vec{B}|^2 \qquad (1)$$

where $\chi_{eff}$ is the effective magnetic susceptibility of the beads, $V$ is the bead volume, $\mu$ and $\mu_0$ are the vacuum and material magnetic permittivity, and $\vec{B}$ is the magnetic field. The magnetic force increases as the gradient of the field squared or twice the field multiplied by the field gradient. Thus, a high magnetic field gradient, not just a high magnetic field, is the key to achieving high bead recovery. Such high gradients can be introduced with sharp geometries (wedges and cones). This geometric enhancement of electromagnetic field has been applied to a variety of engineering designs, from large-scale antenna[28,29] to nanostructures[30,31]. Previously, we used the singular electric field at the edge of microchannels[23] and nanopores[24] to trap colloids and translocate molecules by dielectrophoresis. As shown in Fig. 2a, c, the edge of the nanopore on the sputtered membrane is smooth. The NiFe layer only covered the top of the Au layer due

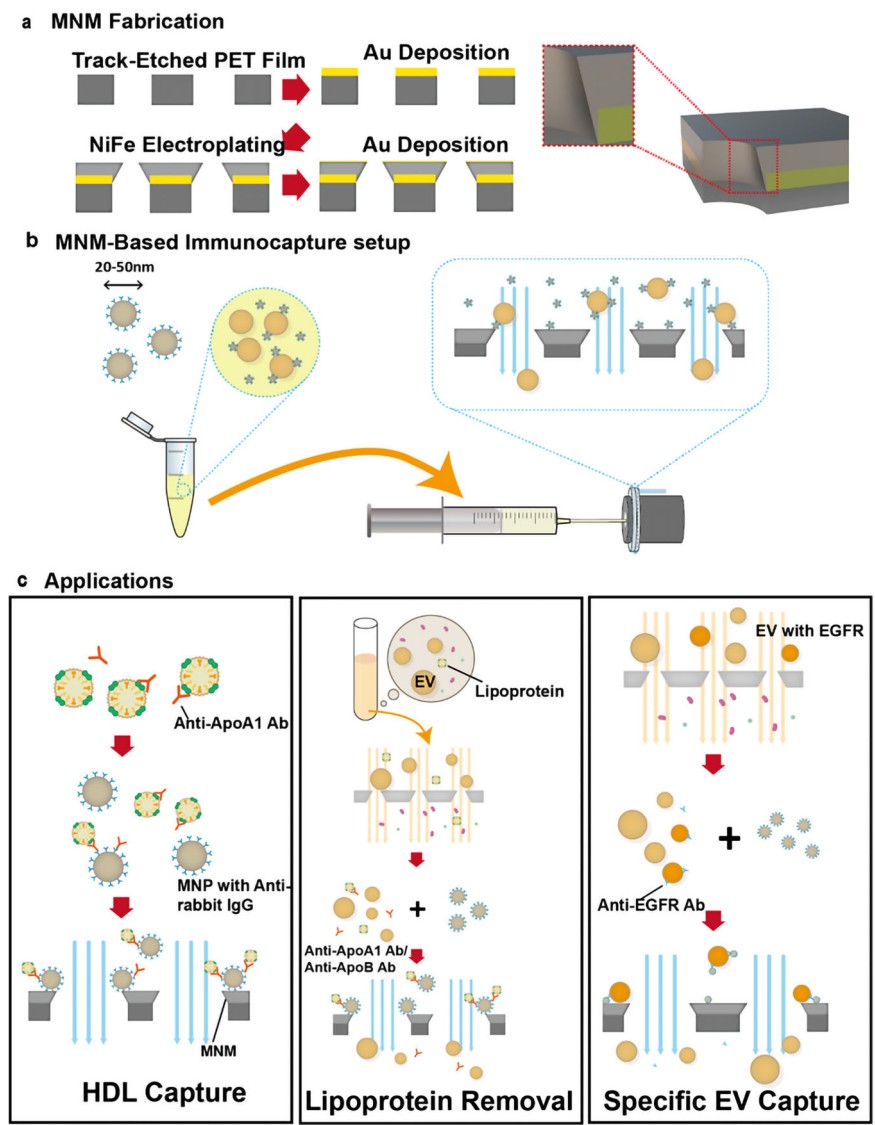

**Fig. 1 Schematic of the experimental procedure. a** Fabrication of the magnetic nanoporous membrane (MNM). In all, 80 nm Au was deposited onto the tracked-etched PET film to provide good adhesion and electric conductivity for electroplating. Then 200 nm NiFe film was deposited onto the membrane with electroplating. In all, 10 nm Au was deposited finally to reduce non-specific adsorption and chemical instability. On the right: The heterogeneous nanoedge junction at the edge of the nanopore on the membrane is highlighted. **b** MNM-based immunocapture setup. First, the antibody and antigen were incubated to form an Ab–Ag complex, followed by incubating with magnetic nanobeads, which were conjugated with anti-rabbit IgG antibodies. Then the diluted sample was run through the chamber of the MNM device with a syringe and pump. The MNM device was assembled with the MNM sandwiched between two 3D-printed chips. The device was assembled between two magnets, with the magnet near the inlet in a ring shape. The magnetic beads were captured onto the edge of the nanopores, as highlighted. **c** Experimental steps of the three applications. Anti-ApoA1 antibodies were used to capture HDL; an Asymmetric nanopore membrane was used to isolate the EVs, with a small amount of HDL remaining, and we used MNM to remove the residual HDL to purify the EV fraction; In the EV fraction, MNM was utilized to capture the specific EV with particular surface protein, i.e., EGFR.

to the anisotropic nature of sputtering. A sharper edge appeared in the electroplated membrane because of the electric field focusing during plating (Fig. 2b, d). The geometric difference can also be observed from the top of the membranes (Fig. 2e, f). Finer crystallization of the electroplated membrane (Fig. 2f) is observed, compared to the sputtered membrane (Fig. 2e), suggesting a sharper corner and higher field gradient can be achieved with the electroplated membrane. The NiFe film also grew inside the pore to form the wedge heterogeneous junction since, unlike sputtering, electroplating also occurs on the side of the 80-nm gold film. Under uniform external magnetization at 0.4 Tesla, a maximum field of 0.62 Tesla and a maximum gradient of flux density square at $2.3 \times 10^5$ T²/cm develops in the water phase (Fig. 2h), compared to 0.48 Tesla and $2.2 \times 10^4$ T²/cm for the sputtered NiFe

film without the wedge ring, which represents a tenfold increase in the force field of Eq. (1).

The high-field enhancement originates from a water phase monopole at the upper edge of the NiFe film, where the wedge angle on the high permeability superparamagnetic NiFe phase is approximately $\pi/2$, and a dipole in the NiFe phase at the outer edge at the base of the wedge junction, where the wedge angle on the NiFe side is ~$3\pi/2$. There is an additional amplification of the dipole field as it enters into the water phase with a magnetic permeability that is 40 times lower. In the sputtered membrane, we only have a weak upper monopole due to the smooth edge (Fig. 2g). This combination of the dipole and monopole field at the sharp edges of the electroplated NiFe film is responsible for the $10 \times$ increase in the nanobead trapping force, and we expect

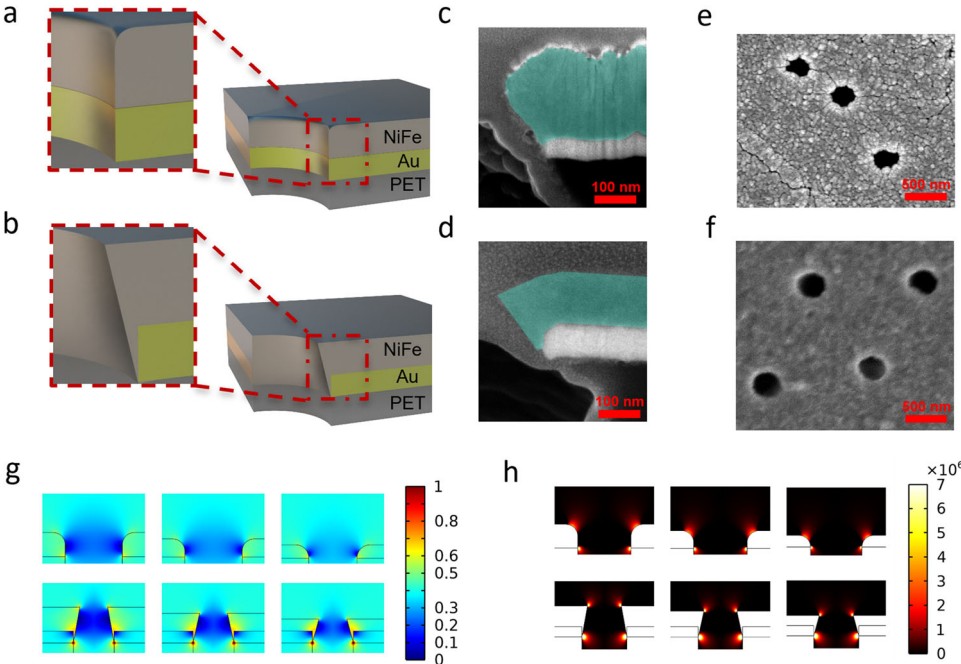

**Fig. 2 Characterization and simulation of the nanopores on the sputtered and electroplated MNM.** Schematic of the nanopores on (**a**) a sputtered magnetic nanoporous membrane, and (**b**) an electroplated magnetic nanoporous membrane, and the edge of the nanopore is highlighted (the final thin gold layer is ignored). **c** Cross-section SEM images of a single nanopore on a sputtered magnetic nanoporous membrane. Note the smooth edge of the NiFe layer (blue). **d** Cross-section SEM images of a single nanopore on an electroplated magnetic nanoporous membrane. Note the sharp edge of the NiFe layer (blue). SEM images of the (**e**) sputtered membrane and (**f**) electroplated membrane. **g** Simulation of magnetic flux density (T) in nanopores on ideal sputtered magnetic nanoporous membrane (upper) and ideal electroplated magnetic nanoporous membranes (lower), showing the dramatic amplification of the flux density at the wedge on the electroplated MNM (membrane thickness left to right: 200 nm, 150 nm, 100 nm). **h** Simulation of the magnetic field, showing the gradient of flux density norm square ($T^2$/cm) at the wedge on both ideal sputtered magnetic nanoporous membranes (upper) and ideal electroplated magnetic nanoporous membranes (lower) (membrane thickness left to right: 200 nm, 150 nm, 100 nm).

to observe a similar increase in capture yield for superparamagnetic nanobeads.

**High-efficiency capture of superparamagnetic beads by MNM.** To test the capture efficiency of the MNM with the multi-edge superparamagnetic wedge around each nanopore, we designed a housing apparatus for MNM immuno-capture applications, which is described in Supplementary Note 3. Round membranes with 2 cm diameter were tested during the experiments. Briefly, 1 mL of $10\times$ diluted 30-nm nanobeads from Exosome Isolation Kit Pan (mouse, Miltenyi Biotec) were passed through the electroplated 450 nm (PET pore size) nanoporous membrane at 1 mL/h. For all experiments, the amount of nanobeads is below the saturation level of the MNM. Figure 3b shows the bead solution before and after magnetic capturing on the membrane; the brownish bead color disappears entirely in the flow-through solution, indicating high bead capture efficiency. Nanobeads convected by streamlines close to the surface are trapped by the monopole near the top edge of the NiFe film, as shown in Fig. 3a. The remaining beads are convected into the pore center and are trapped by the high-dipole magnetic force within the pore (Fig. 3c). Healthy mouse plasma was used to validate the EV capture ability of the MNM (details in Supplementary Note 4). SEM image (Fig. 3d) shows mouse plasma exosome docked with nanobeads captured near the nanopore.

The pore size shrinks from 450 to about 350 nm after electroplating, which is still larger than the typical small EV (sEV) size of 30–200 nm, allowing non-target EVs without nanobeads to pass through. A quantitative study of bead capturing efficiency was conducted by comparing the bead concentration measured by nanoparticle tracking analysis (NTA)

before and after capturing (see Fig. 3e). At 1 mL/h, >99% of the beads were captured. The bead capture efficiency did not diminish even at a flow rate of 5 mL/h. This throughput is high enough for most extracellular vesicle immunocapture applications. Furthermore, only 13% of the beads were lost when the flow rate was increased to 10 mL/h. For membranes with 1-μm pore size, the bead capture efficiency was still >80% at 1 mL/h. In stark contrast, only 22% of beads were captured by the sputtered 450-nm membrane (Fig. 3e). For larger vesicles above 300 nm, electroplated MNM with 1μm pore size can be used at a lower flow rate or with a higher external magnetic field.

**Isolation of high-density lipoprotein (HDL).** Based on the effectiveness of the MNM in capturing EVs, we sought to investigate the capacity of this method to capture lipoproteins, namely HDLs (Fig. 4a). HDLs are highly abundant in plasma and other biofluids and provide a good model to trace based on standard cholesterol assays that can be used to quantify them. Apolipoprotein A-I (apoA-I) is the main structure–function protein on the surface of HDL particles. ApoA-I is primarily associated with HDL and accounts for ~70% of total HDL protein content by mass. HDL samples were isolated from human plasma by density-gradient ultracentrifugation (DGUC), and total protein levels were quantified by colorimetric assays[32]. For capture, 2 μg anti-ApoA-I (Abcam, ab52945, rabbit monoclonal to ApoA1) antibodies were mixed with 100 μL of 100 μg/mL HDL sample, incubated for 30 min, and treated with 100 μL anti-rabbit IgG nanobeads (30 nm, Milyteni Biotec) for 1 h. After the HDL was immuno-captured by the magnetic nanobeads, the solution was diluted to 500 μL with $1\times$ PBS and passed through the 450-nm electroplated MNM membrane, followed by flushing with

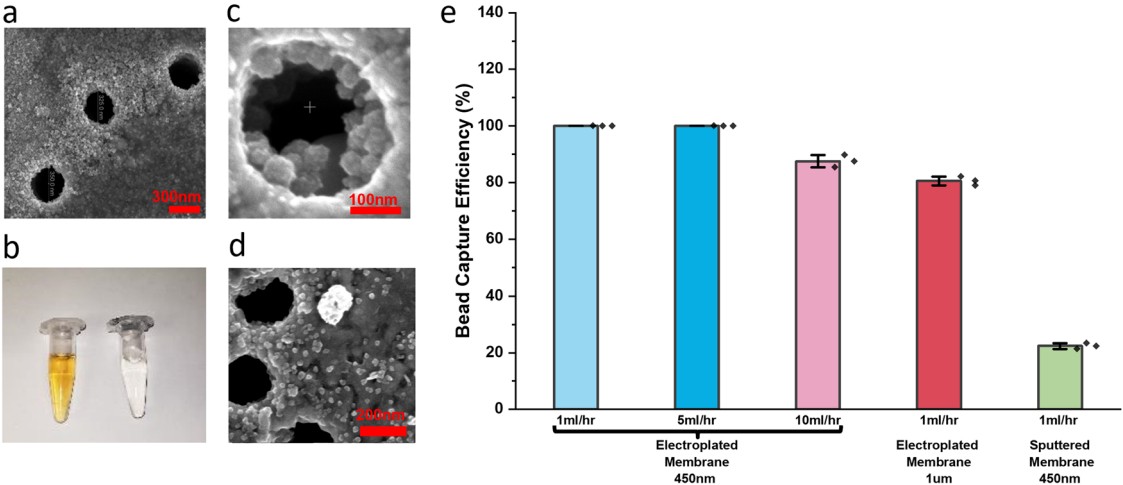

**Fig. 3 Characterization of the magnetic beads capture of MNM. a** SEM image of captured magnetic beads near the edges of nanopores by the monopole field. The diameter of the pore has decreased to around 300 nm after the deposition of different metallic layers. **b** Solution of beads before passing through the magnetic nanoporous membrane (left) and after filtering through the magnetized membrane (right). The yellow color indicates the concentrated beads. **c** Zoomed-in SEM image shows nanobeads captured inside a single nanopore by the dipole field of the wedge junction. **d** SEM image of mouse plasma exosome captured near the nanopore. **e** Bead capture efficiency of sputtered and electroplated membranes at different flow rates and pore sizes. For the electroplated membrane, the original pore size of 450 nm and 1 μm and flow rates of 1, 5, and 10 mL/h have been tested. For the sputtered membrane, the original pore size of 450 nm and flow rate of 1 mL/h have been tested ($n = 3$ independent experiments). Error bars indicate the standard deviations (SD) for each condition.

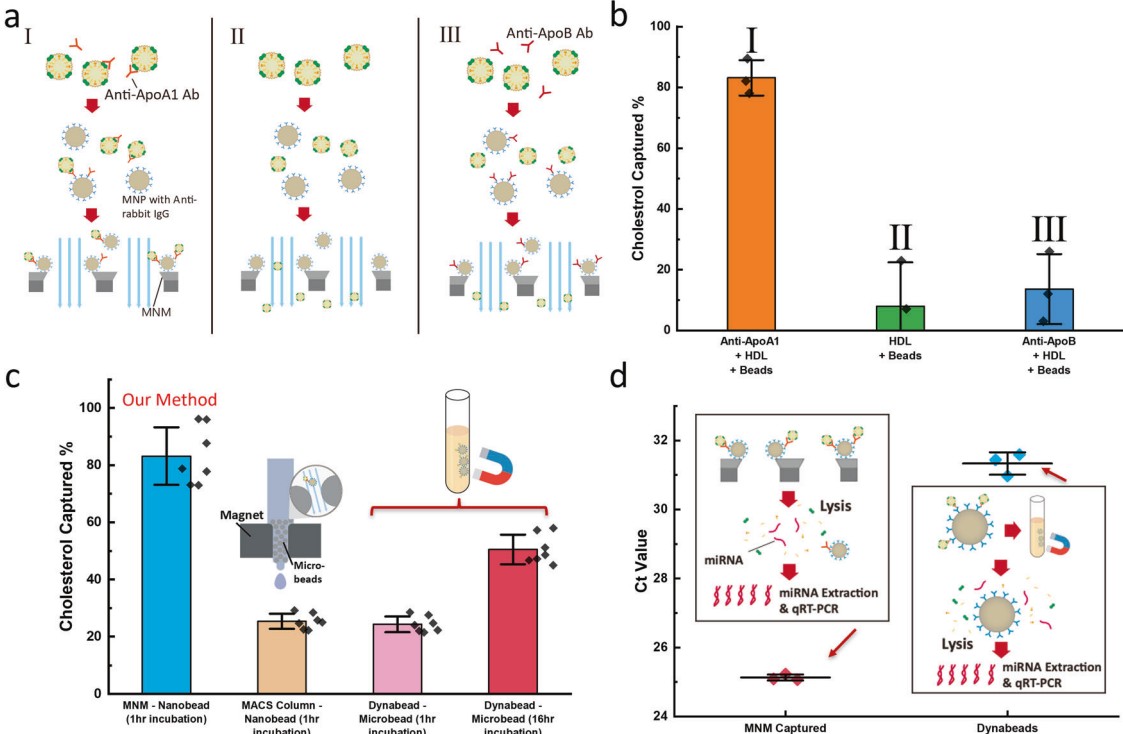

**Fig. 4 Characterization of the HDL capture rate. a** Schematic of the immunocapture of HDL capture rate using cholesterol as a measure. In the three cases, different combinations were tested for specific capture of HDL with anti-ApoA1 antibodies and non-specific capture with no antibodies and anti-ApoB antibodies, which are specific to LDL. **b** Capture rate of the three cases in (**a**) ($n = 3$ measurements in each case). **c** Comparison of HDL capture rate using different immunocapture kits, including the Miltenyi MACS™ μColumn and Thermofisher Dynabeads™. The inset schematics show the basic working principle of different technologies. The incubation time was chosen to be 1 h and 16 h for Dynabeads™ ($n = 7$ measurements for each method). **d** Ct value of miR-21 from qRT-PCR experiments. MiRNA samples were extracted from HDL captured by both MNM and Dynabeads™ with the same starting sample volume ($n = 3$ measurements for each sample). The incubation time of the MNM experiment is 1 h, and that of Dynabeads™ is 12 h. The schematics of the immunocapture and qRT-PCR were shown in the inset. The HDL was captured by MNM or Dynabeads and then lysed, followed by miRNA extraction and qRT-PCR. Error bars indicate the standard deviation (SD) in each plot.

1 mL 1 × PBS to bring all beads onto the membrane surface and to remove the residual HDL solution in the chamber. The flow-through was collected for each sample. Since cholesterol is present only in the lipoproteins and not the EVs, the concentration of cholesterol was measured to calculate the total amount of cholesterol in both the original sample and flow-through (Fig. 4b). The cholesterol capture rate can be calculated as follows:

$$Cholesterol\ Captured\% = \frac{Cholesterol_{original} - Cholesterol_{flow-through}}{Cholesterol_{original}} \times 100\% \quad (2)$$

Remarkably, >80% of HDL was recovered using this approach. To confirm the specificity of the immunocapture and non-specific adsorption in our device, two negative controls were tested. If no antibodies were functionalized onto the nanobeads in the experiments, <10% of HDL was lost in the device, which was due to non-specific adsorption and experimental error. When antibodies (Abcam, ab139401, rabbit monoclonal to ApoB) against apolipoprotein B, the structural protein for low-density lipoproteins (LDL), were used instead of anti-ApoA-I, the loss increased to 14%. The additional 4% loss may come from the non-specific capture of HDL by anti-ApoB. In both negative controls, the non-specific capture rate of <15% is significantly lower than the specific capture rate of 80%. We benchmarked our method to a commercial immunocapture kit using their standard protocol (see Supplementary Note 5). As shown in Fig. 4c, for nanobeads, only 20% HDL was captured by the μColumn (Milyteni Biotec) because of the low bead capture efficiency of the packed column. Furthermore, for microbeads like Dynabeads™, even after 16 h of incubation, which is much longer than the standard protocol, the HDL capture efficiency does not exceed 50%. For the same incubation time of 1 h as the nanobeads, only 25% HDL was recovered (Supplementary Fig. S4), marginally >15% non-specific capture rate.

To further demonstrate the advantage of the MNM immuno-capturing method, miRNA extraction and qRT-PCR quantification of miR-21 were performed on HDL captured by both MNM and Dynabeads™. Figure 4d shows a Ct difference of more than 6 between the two immunocapture methods, which suggests the miRNA expression result of Dynabeads™ is 64-fold lower than that of MNM (delta-delta Ct method). The long incubation time required by microbeads causes sample degradation and miRNA degradation, and adsorption, which leads to significant bias in miRNA quantification. In contrast, a high concentration of miR-21 was preserved in the fast MNM immunocapture.

**Purification of EVs in filtered plasma**. In this demonstration, 100 μL healthy human plasma was first diluted and processed by tangential flow filtration with 30 nm asymmetric nanoporous membranes[33] to remove most of the HDL and other lipoproteins (Fig. 5a). As shown in Fig. 5b, there was still 17% cholesterol left from mostly LDL and VLDL after filtration. A small amount of HDL could also be present in the filtered sample due to the dominant amount of HDL in the original plasma. Therefore, we mixed the filtered sample with 2 μg anti-ApoA-I and 2 μg anti-ApoB antibodies and incubated it for 30 min. These apolipo-proteins are specific to the lipoproteins but not EVs. After adding 200 μL anti-rabbit IgG microBeads (30 nm, Milyteni Biotec) and incubating for 1 h, the mixture was passed through the MNM. The collected flow-through was the purified EV sample, which contained 85% of the original EV from NTA characterization in Fig. 5c, but only 5% of the original cholesterol. The size distribution of the EV sample was also preserved after purification, as seen in Fig. 5c, indicating that MNM not only retains 85% of EVs in number but also avoided EV lysing or coalescence. ELISA results for CD63 and CD9 in Fig. 5c also confirmed a minimal

loss (13–17%) before and after MNM purification with anti-ApoA-I and anti-ApoB pull-down procedures.

**Isolation and purification of EGFR EVs from DiFi cell lines**. A major research direction in the EV field is to identify EVs secreted from specific (diseased) cells or by specific pathways[14]. In this study, EVs were first isolated from human colorectal cancer cells (DiFi) by a size-based ANM (asymmetrical nanoporous membrane) separation technology. Then specific EVs with EGFR membrane proteins from the ANM EV isolate were further isolated by the MNM to demonstrate accurate miRNA quantification of a specific subclass of EVs (Fig. 6a). Based on analysis of EGFR-containing DiFi EVs, isolated EVs were likely exosomes based on tetraspanin content[34]. Some of the EVs released by the DiFi cells exhibited inactive EGFR and active EGFR[34]. We utilized a total EGFR antibody that captures both active and inactive EGFR in this experiment. DiFi cell culture supernatants were first processed by tangential flow filtration with 30-nm asymmetric nanoporous membranes[33] to remove free-floating proteins. Briefly, 1 μg anti-EGFR antibodies were added to the sample and incubated for 30 min. After adding 100 μL anti-human IgG microBeads (30 nm, Milyteni Biotec) and incubating for 1 h, the mixture was passed through the MNM. Western blots of syntenin-1 (as EV protein[15,35]) and EGFR are carried out on both ANM-isolated fraction and MNM captured fraction to validate the EV content (Fig. 6e). We extracted miRNA from all fractions during the process and performed qRT-PCR to assess miR-21 levels. Figure 6b shows miRNA content inside the isolated EGFR EVs, and the flow-through material adds up equally to the total miR-21 levels in the original sample with a 21% error, which is insignificant considering qRT-PCR can only differentiate twofold changes. The total amount of EGFR in the ANM isolate, and MNM flow-through is also measured by ELISA. A drop of close to 90% was achieved, indicating most EGFRs were captured in the MNM EGFR isolate (Fig. 6b, inset). The CD63 and CD9 concentrations inside the MNM EGFR isolate are close to half of the total before MNM capture (Fig. 6c), which is consistent with the miR-21 qRT-PCR result (Fig. 6b). To further explore the quantification potential of our system, we did the same experiment on both undiluted and 8 × diluted ANM-processed DiFi samples. As shown in Fig. 6d, the 8.3-fold change in miR-21 expression level matches the dilution factor, suggesting our high efficiency is consistent among different initial sample concentrations, which is important for quantitative biomarker studies.

**Discussion**
Here, we demonstrated the utility and efficiency of electroplated MNM with unique heterogeneous superparamagnetic junctions. This method can achieve high-efficiency capture of super-paramagnetic nanobeads. We achieved almost 100% nanobead recovery from the solution at up to 5 mL/h on a single device. The uncaptured EVs can go through the straight pores and be collected in the flow-through. We proved our device's efficiency and specificity using HDL as a model, with >80% of HDL particles recovered and minimal non-specific retention at less than 15%. The high and consistent yield of our system provides quantification potential for studies of EVs, lipoproteins, and other extracellular RNA carriers. We further demonstrated the performance of MNM in exosome capture, purification of HDL-enriched EV samples, and EGFR-positive EVs characterization. The direct lysis protocol in this study is applicable to a variety of downstream analyses for biomarker discovery and diagnostics, such as qRT-PCR, MS-based proteomics, sequencing, etc. For drug delivery, tissue engineering, and other applications requiring intact EVs as carriers, an EV-releasing protocol is necessary.

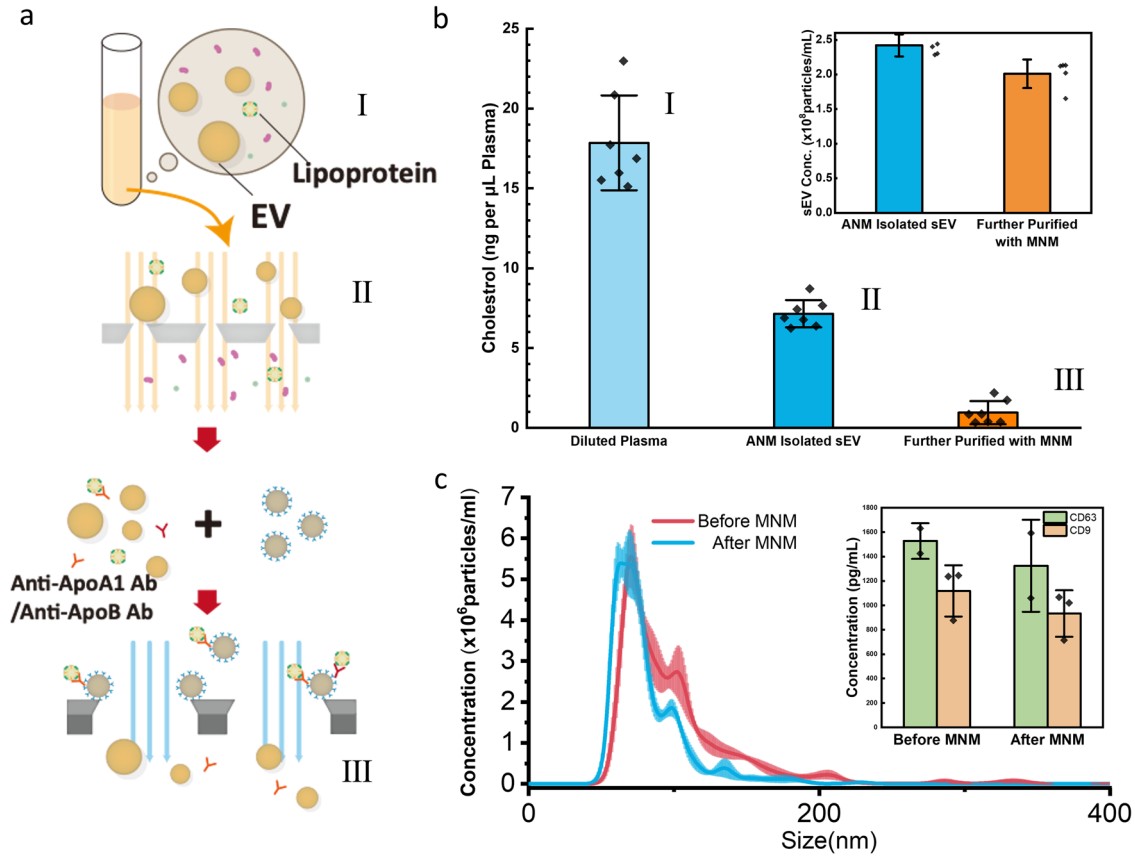

**Fig. 5 Characterization of the HDL removal from the fractionated EV and yield/quality of the isolated EV. a** Schematic of the EV fractionation and immunocapture of HDL. The diluted plasma (sample I) runs through the asymmetric nanoporous membrane to remove other particles with a size-exclusion mechanism, thus obtaining the EV fraction (sample II). Due to the dominating amount of HDL, a small portion of HDL remained in the EV fraction, so it was incubated with anti-ApoA1 and anti-ApoB antibodies, and then magnetic nanobeads conjugated with anti-rabbit IgG antibodies and run through the MNM device to remove the HDL, and the flow-through was collected (sample III). **b** Lipoproteins (cholesterol) remnant at different stages of purification from samples I, II, III in **a**), and (inset) EV concentration before and after MNM immunocapture (sample II and III) ($n = 5$ measurements for each sample). **c** Size distribution of EV samples with NTA before (sample II) and after MNM immunocapture (sample III) showing minimal EV loss or lysis/coalescence. (inset) CD63 and CD9 concentration before and after MNM immunocapture (sample II and III) measured by ELISA ($n = 2$ and 3 measurements for CD63, CD9, respectively). Error bars indicate the standard deviation (SD) in each plot.

Dissociation buffers[36,37], photo-cleavable linker[38], and protease-sensitive linkers[39] are potential strategies to detach the EVs from the MNM. Our platform is also applicable for other molecular or virus immunocapture applications where capture efficiency and throughput are essential. In addition to the isolation of specific EVs from plasma for liquid biopsy applications (disease screening and therapy management), the MNM technology should also be useful for biomarker discovery from cell cultures, as we have demonstrated here for the DiFi sample, and also for organ-on-the-chip or organoid models[35,40–42].

## Methods

**Numerical simulations.** COMSOL was used to model and simulate different nanopore structures to estimate the magnetic flux density and its gradient. A two-dimensional (2D) axial-symmetry geometry model was used with the Magnetic Fields, No Currents interface in the AC/DC module. The software built-in NiFe B-H curve was used. A static magnetic flux density of 0.5 T was applied at the far boundary of the model. The simulation was conducted with a physics-controlled meshing of extremely fine elements. More details have been shown in Supplementary Note 1.

**Microscopy imaging.** Surface SEM images were taken with Magellan 400. For EV-captured membranes, a 2% EMS-quality paraformaldehyde aqueous solution was used for fixation, and 2-nm gold was sputtered in advance for conductivity. Vesicles were examined under low beam energies. Cross-sections of the nanopores were prepared using the Helios G4 UX DualBeam (Thermo Scientific). After protecting the cross-section surface with Pt EBID, slices of 5-nm thickness were

sequentially obtained with Auto Slice & View™ 4 (AS&V4) software operating with a focused 10 keV beam of gallium ions. The slicing was stopped at the center of the pore, and the images were acquired with a voltage of 3 kV using a TLD detector for secondary electrons.

**$Ni_{80}Fe_{20}$ deposition by electroplating.** The used track-etched PET films (PET115745, Wuwei Kejin Xinfa) are 11-μm thick and have a pore density of $5 \times 10^7/cm^2$. To fabricate the electroplated magnetic nanoporous membrane, 80 nm Au was deposited onto the track-etched PET films in an FC-1800 Evaporator. The gold layer provides good adhesiveness between polymer and NiFe and functions as a seed layer for electroplating. The membrane was cut into 4 cm × 4 cm pieces. Copper tapes were used to fix membranes onto the support and electrically connected to the cathode. A nickel plate was used as the anode. The electroplating solution adapted from the literature[43,44] can be found in Supplementary Note 6. Constant current density at 2 mA/cm² was applied by Keithley 2636A Dual-Channel System SourceMeter; voltage is monitored during the electroplating process. A custom electroplating stirring tank was designed for uniform deposition. The deposition rate was derived by SEM images on thicker samples grown under the same conditions. Another 10-nm Au was deposited on the top of the NiFe layer to reduce non-specific adsorption and chemical instability. Additional characterizations of the membranes are detailed in Supplementary Note 2.

**$Ni_{80}Fe_{20}$ deposition by sputtering.** Same as the electroplated samples, 80-nm Au was deposited onto the PET films initially. The sputtered samples were prepared at room temperature in a commercial UHV sputtering system Oerlikon DCSS using a $Ni_{80}Fe_{20}$ target. Ar gas flow was fixed to 20sccm, and the plasma power was 50 W during deposition. The deposition rate was derived by means of a stylus profilometer and SEM images on thicker samples grown under the same conditions. After sputtering, 10 nm Au was deposited at the top of the NiFe layer.

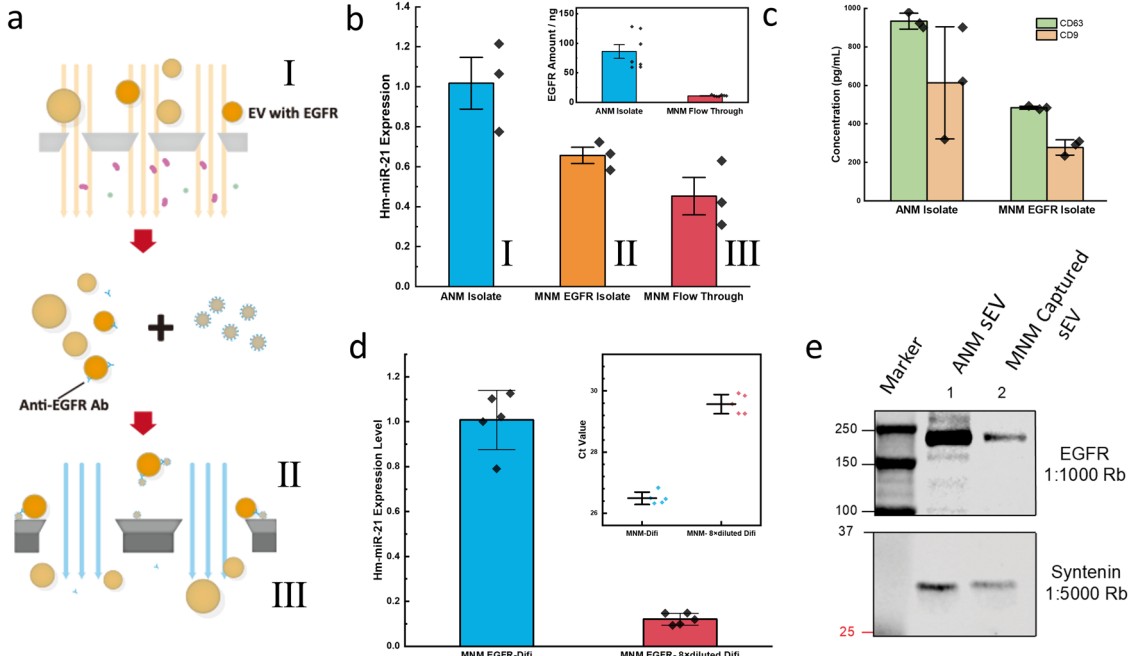

**Fig. 6 Characterization of MNM immunocapture of specific EV, i.e., EV with EGFR. a** Schematic of the EV fractionation and immunocapture of EV with EGFR. The DiFi cell culture medium was run through the asymmetric nanoporous membrane to obtain the EV fraction by size separation (sample I), which was then incubated with the anti-EGFR antibodies and then magnetic nanobeads conjugated with anti-human IgG antibodies. Then the samples passed through the MNM device, with the captured sample on the magnetic membrane mixed with lysing buffer (sample II), and the flow-through was collected (sample III). **b** Hm-miR-21 expression level of different DiFi exosome fractions from samples I, II, III in (**a**) ($n = 3$ measurements for each sample). (inlet) The total amount of EGFR was measured for I and III, indicating most EGFR-positive EVs from the ANM-isolated EVs are captured by MNM ($n = 7$ measurements for each sample). **c** CD63 and CD9 concentration of ANM isolate (sample I) and MNM EGFR isolate (sample II) measured by ELISA. ($n = 3$ measurements for each sample) **d** Expression level of Hm-miR-21 in the EGFR exosomes before and after 8 × dilution of the DiFi samples and the Ct value of their qRT-PCR results ($n = 5$ measurements for each sample). **e** Western blot of the isolated EVs from the DiFi cell line. After the Marker Lane then (1) ANM-isolated sEV; (2) MNM bead captured sEV. Each well was loaded with 40 μg protein. Odyssey exposure (bottom: 10 min, top: overexposed). The top panel is blotted by EGFR and the bottom by syntenin-1, respectively. Error bars indicate the standard deviation (SD) in each plot.

**Plasma samples.** De-identified plasma samples were obtained from Zen-Bio Inc. and consisted of 10 mL of fresh human plasma collected in tubes with EDTA coagulant. Each sample was tested for pathogens as required by the FDA. All assay protocols performed in studies involving human participants were in accordance with the ethical standards of the University of Notre Dame.

**DiFi cell culture-conditioned media collection.** DiFi cells were grown in a C2011 FiberCell bioreactor with 20 kDa pore using the manufacturer's instructions (FiberCell Systems, New Market, MD) using FiberCell systems' defined serum-free media (CDM-HD). Specifically, the bioreactor was washed overnight with sterile 1 × DPBS (Corning, Corning, NY) and then overnight with high glucose DMEM (hgDMEM/ Corning). The bioreactor was treated with 0.5 mg of bovine fibronectin (Sigma, St. Louis, MO) in 20 ml of DMEM for 4 h to overnight. The bioreactor was then washed overnight with complete hgDMEM with 10% bovine growth serum (1% penicillin–streptomycin [Pen/Strep, GIBCO, Dublin/Ireland], 1% glutamine [GIBCO], 1% glutamine [GIBCO], 1% nonessential amino acids [GIBCO]). The bioreactor was loaded with $1-5 \times 10^8$ DiFi cells in complete hgDMEM with 10% serum and allowed to stand for 1 h before circulating complete DMEM with 10% serum. Glucose levels were monitored daily with a glucometer (CESCO bioengineering, Trevose, PA), and when glucose levels were at half of that in starting media, the media bottle was replaced. In subsequent media changes, the bioreactor went from 10% bovine serum to 5% then to 3%, before switching to 10% CDM-HD (DMEM-HD) media. Once cells were established in DMEM-HD (at least 2 weeks in DMEM-HD), a routine harvest of conditioned media was performed, removing 20 ml of conditioned media per day. Collected media was spun at 2000 rpm to remove cells and any large debris, then a subfraction of the media was additionally gravity filtered through a Millex 0.22-μm pore syringe filter (Millipore Sigma, Burlington, MA). At least 3 days of filtered media collections were pooled.

**Lipoprotein collection.** Plasma was collected from consented human participants under active Vanderbilt IRB protocols and guidance. Blood was drawn into EDTA-containing collection tubes and immediately centrifuged to separate plasma. HDL and LDL were isolated from human plasma by KBr density-gradient ultra-centrifugation (DGUC), as previously described[32]. Briefly, native LDL (1.019–1.062 g/L) and HDL (1.063–1.021 g/L) were isolated by sequential DGUC

using an Optima XPN-80 Ultracentrifuge with SW41Ti or SW32Ti rotors (Beckman–Coulter). HDL and LDL were dialyzed in PBS with >4 buffer changes and concentrated with 3000 Da m.w. cutoff filters (Millipore). Total protein levels were determined for each lipoprotein sample (HDL and LDL) by BCA colorimetric assays (Pierce, ThermoFisher).

**Magnetic nanobeads.** Magnetic nanobeads are purchased from Miltenyi and used as is. These nanobeads are 20–30 nm (checked with SEM) and functionalized with antibodies. Exosome Isolation Kit Pan, mouse (Cat#130-117-039), anti-rabbit IgG MicroBeads (Cat#130-048-602), and Anti-IgG MicroBeads, human (Cat#130-047-501) are used respectively for each experiment.

**Cholesterol assay.** Cholesterol Quantification Assay Kit (Sigma-Aldrich, CS0005) was used to measure the cholesterol concentration of samples. Briefly, 44 μL Assay Buffer, 2 μL Probe, 2 μL Enzyme Mix, 2 μL Cholesterol Esterase, and 50 μL sample were mixed and incubated at 37 °C for 30 min in each well. A calibration curve was established for every measurement with standard samples with 0–5 μg cholesterol. All samples were diluted to the range of the calibration curve with the Assay Buffer. Absorbance at 570 nm was measured and compared to the standards on the same plate to determine total cholesterol.

**qRT-PCR.** miRNAs were isolated from samples using the NucleoSpin® miRNA Plasma Kit (Takara Bio) according to the manufacturer's manual. 300 μL of the sample was first mixed with 90 μL MLP solution and incubated at room temperature for 3 min, followed by adding 30 μL MPP buffer and 1 min room temperature incubation. 3.5 μL ($1.6 \times 10^8$ copies/μL) of cel-miR-39-3p in RNase-free water was added into the lysate as a normalization spiked-in control. Then the mixture was centrifuged at 11,000 ×$g$. The supernatant was taken and mixed with 400 μL iso-propanol. The mixture was transferred into the binding column and centrifuged at 11,000 × $g$ for 30 s. The column was then washed with 100 μL MW1 and 700 μL MW2 sequentially at 11,000 × $g$ for 30 s, followed by 250 μL MW2 washing and drying at 11,000 × $g$ for 3 min. Finally, 30 μL RNase-free water was added to elute the miRNA at 11,000 × $g$ for 1 min after incubation at room temperature for 1 min. Reverse transcription was carried out using a miScript II RT Kit (Qiagen). A 20 μL reverse transcription reaction was prepared with 2.2 μL of eluted miRNA, 4 μL 5 ×

miScript HiSpec Buffer (Qiagen), 2 μL 10 × miScript Nucleics Mix (Qiagen), 9.8 μL RNase-free water, and 2 μL miScript Reverse Transcriptase Mix (Qiagen). The reaction was incubated at 16 °C for 60 min, followed by 95 °C for 5 min. The reverse transcription reaction was then diluted with 200 μL RNase-free water. Triplicates of qPCR reactions were carried out using miScript SYBR Green PCR Kit (Qiagen) and run on a StepOnePlus™ Real-Time PCR System (Applied Biosystems). The reaction contained 2 μL diluted cDNA, 12.5 μL 2· QuantiTect® SYBR Green PCR Master Mix (Qiagen), 2.5 μL 10· miScript Universal Primer (Qiagen), 10· miScript Primer Assay (Qiagen) for the target miRNA, and 5.5 μL RNase-free water in a final volume of 25 μL. The reaction mixtures were incubated for 15 min at 95 °C, followed by 45 cycles of 94 °C for 15 s, 55 °C for 30 s, and 70 °C for 30 s. The Ct values were acquired and analyzed using StepOne™ Software v2.3 in accordance with the MIQE guidelines[45]. The Ct values of the target miRNAs were adjusted by spiked-in standard control (cel-miR-39-3p) added during miRNA extraction. The expression level is calculated by the delta–delta Ct method.

**ELISA**. Human EGFR ELISA kit (EGFR0, R&D Systems™), Human CD63 ELISA Kit (Cat#EH95RB, Invitrogen), and Human CD9 ELISA Kit (#MBS7607059, MyBioSource) were used to quantify specific proteins in the samples respectively according to the manufacturer's instructions. EV markers CD9 and CD63 selected from MISEV2018 guidelines[46] were detected in EV samples to varying concentrations. Standard curves were established for each plate, and the concentrations of proteins were determined by the readings.

**Western blots**. Western blots were done according to the general protocol described previously[47]. Briefly, proteins were quantified by BCA (Thermo, Cat# 23235) using the manufacturer's instructions. Forty micrograms of protein were loaded in each lane of an 11% SDS-poly acrylamide gel and electrophoresed at 160 V for about 5 h. Resolved proteins were transferred to nitrocellulose membrane overnight at 4 °C at 25 volts and then blocked with Intercept blocking buffer (Li-COR, Cat# 927-60001) for 4–5 h. Nitrocellulose membranes were cut into molecular weight regions for blotting based on apparent molecular weight as demarked by size standards (Bio-Rad, Cat# 1610374). EGFR (Millipore Rb, 1:1000, Cat#06-847), and Syntenin (Abcam, Rb, 1:5000, Cat# Ab133267) antibodies were used for the immunoblots. Nitrocellulose was cut into the top, middle and bottom, respectively for these markers. Blots were then probed with secondary Goat anti-rabbit IRDye 800 CW (LI-COR, 1:5000, Cat# 926-32213). Membranes were developed by Odyssey (Li-COR).

**Nanoparticle tracking analysis**. Nanoparticle tracking analysis (NTA) was performed using a NanoSight NS300 (NanoSight Ltd., Amesbury, UK) according to MISEV2018 guidelines[46]. All samples were diluted to the optimal working particle range prior to measurements using 1 × PBS. Five 60 s videos were recorded of each sample with the camera level set at 10. A constant flow rate setting of 1000 was maintained during the recording. The temperature was monitored throughout the measurements. The instrument was flushed with 1 × PBS between measurements. Videos recorded for each sample were analyzed with NTA software to determine the concentration and size distribution of measured particles with corresponding standard error. The same detection threshold was used for analysis.

**Statistics and reproducibility**. The number of biological replicates and measurements made are clarified in each figure. The standard errors of all datasets are calculated and plotted using the software OriginPro and double-checked manually. Data are shown as individual data points and mean ± SE.

**Reporting summary**. Further information on research design is available in the Nature Portfolio Reporting Summary linked to this article.

## Data availability
Source data for graphs and charts can be found in Supplementary Data associated with this article. Additional data that contributed to this study are present in the Supplementary Information. The uncropped and unedited gel images are included in Supplementary Fig. S7. All other data related to this paper may be requested from the corresponding authors.

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

## Acknowledgements

The authors acknowledge the support of the NIH Common Fund through the Office of Strategic Coordination/Office of the NIH Director, 1UH3CA241684-01. C.Z. acknowledges the support of a China Scholarship Council fellowship. R.J.C. acknowledges the support of NCI R35 CA197570. The authors acknowledge the support of the Cancer Cure Venture (CCV) Grant, made possible by the Walther Cancer Foundation.

## Author contributions

C.Z., X.H., C.W., and H.C.C. conceived the idea and designed the study. C.Z. performed the finite element simulations. C.Z., X.H., C.W., and Z.C. performed the experiments. C.Z. analyzed the results and wrote the manuscript with inputs from all authors. Y.Z., L.X., J.N.H., J.L.F, K.C.V., and R.J.C. provided biological samples. C.W., S.S., and H.C.C. supervised the project.

## Competing interests

The authors declare no competing interests.
