## [Peer Review File · Communications Biology]

Reviewers' comments:

Reviewer #1 (Remarks to the Author):

In this report, Zhang et al. developed a novel magnetic nanoporous membrane (MNM) with a unique superparamagnetic nano edge ring. They further use high-density lipoproteins (HDL) sample, filtered plasma sample and culture supernatant of human colorectal cancer cells (DiFi) to test the capture ability. Overall, the topic is interesting because it provides new methods for the immunocapture of specific molecular nanocarriers, particularly for the Extracellular Vesicles (EVs) . However, there are some concerns about the experiments and data.

The characterization of the isolated EVs is very poor in 2.4 and 2.5. No characterization of the total EVs and the specific EVs, only use qRT-PCR, ELISA and NTA to simply characterization of the isolated sample. I am not sure whether the isolated sample is EV or not. I think it is necessary to make full detailed analyses of the isolated sample step by step

In Figure4, Cholesterol was decreased after MNM purification (about loss 6/7 after MNM purification), but the number of EVs did not significantly decrease. Why, is the plasma have high level of LDL cholesterol?

Figure 5, EGFR protein is notably decreased after MNM purification compared with ANM isolate. Why?

Minor:

1. What are the HDL samples? I can't find the information.

Reviewer #2 (Remarks to the Author):

This paper demonstrated the utility and efficiency of the electroplated magnetic nanoporous membrane (MNM) with unique heterogeneous superparamagnetic junctions and this method could achieve high-efficiency capture of superparamagnetic nanobeads and quantify biomarkers carried by EVs and lipoproteins in heterogeneous physiological fluids.

The theme is of significance and helps researchers in biomedical fields to further enhance the capture efficiency of specific extracellular vesicles and lipoproteins. The content of this article is comprehensive and the skeleton is clear. Thus, I recommend the acceptance of this manuscript in Communications Biology after the following revisions:

1. The nanobeads used in this study should be introduced briefly to provide more information.
2. Figures 1c and 1d showed the cross-section SEM images of a single nanopore on a sputtered magnetic nanoporous membrane and a single nanopore on the electroplated magnetic nanoporous membrane. It is recommended to reduce the magnification to get an overview of SEM images. The structure of the nanopore edge could be exhibited better if characterized by elemental analysis than highlighted by the blue color block.
3. As Figure 2c shows, the nanobeads were captured inside a single nanopore by the dipole field of the wedge junction. When the sample was transported at high flux for a long time, could the nano-magnetic beads block the nanopores, and thereby reduce the capture efficiency and the yield of specific isolation?
4. The reviewer is wondering whether the extracellular vesicles and lipoproteins captured by MNM can be re-released for subsequent applications.
5. To make the comparison of the EVs size distribution before and after purification more intuitively, it is suggested to combine Figures 4c and 4d.
6. There are a lot of minor problems in the figures and should be corrected carefully. Figure 1c is too blurry and some extra borders in other figures should be removed, such as Figures 1g, 1h, 3, 4, and 5.

7. The signs of statistical significance in Figures 2e, 3c, and 5b should be well-positioned. In addition, the scale bar in Figure 2d should be clearer and the picture of the actual experimental setup in Figure S2b could be optimized.

8. Recently, there has been quite a lot of research published on this topic area, especially organ-on-chips. The authors are suggested to read them and use them as references. The following articles may be mentioned:

Science Bulletin 2019, 64, 1110-1117.

Research 2019, 2019, 6906275.

Engineered Regeneration, 2020, 1, 35-50.

Research 2021, 2021, 9829068.

Engineered Regeneration, 2021, 2, 182-194.

Advanced Science 2022, 9, 2104272.

Reviewer #3 (Remarks to the Author):

Zhang et al. designed the utility and efficiency of electroplated magnetic nanoporous membrane with heterogeneous superparamagnetic junctions for immunocapture of HDL particles and specific extracellular vesicles. Generally, the manuscript should be revised in order to make it more accessible to the general audience.

1. Some figures are not described in the results section of the manuscript, such as Figures 1e and f, Figures 2d.

2. The supporting information in the results section should be marked with the corresponding figure number and explained in detail.

3. The Discussion should be expanded.

4. In the figure legend of Figure 2e (Line 516), 10 mL/mL should be corrected to 10 mL/hr. In addition, groups should be marked more clearly in this image, which might easily confuse the reader.

5. In the Figures 4 and 5, it is not enough to use NTA alone to confirm whether the isolated particles are EVs. Transmission electron microscopy images should be supplemented to confirm the morphology of EVs isolated by MNM, and western blot should be used to verify the marker proteins of EVs. In addition, since EGFR is a transmembrane protein, Western blot should be used to verify whether high purity EGFR-specific EVs can be isolated by the MNM.

Responses to Reviewer #1 :

In this report, Zhang et al. developed a novel magnetic nanoporous membrane (MNM) with a unique superparamagnetic nano edge ring. They further use high-density lipoproteins (HDL) sample, filtered plasma sample and culture supernatant of human colorectal cancer cells (DiFi) to test the capture ability. Overall, the topic is interesting because it provides new methods for the immunocapture of specific molecular nanocarriers, particularly for the Extracellular Vesicles (EVs) . However, there are some concerns about the experiments and data.

We are glad that the reviewer finds our technology novel and the paper interesting. We have modified the manuscript to clarify the issues the reviewer brought up and conducted new experiments (with a new coauthor as a consequence) he/she suggested. It is now a much clearer and stronger manuscript.

The characterization of the isolated EVs is very poor in 2.4 and 2.5. No characterization of the total EVs and the specific EVs, only use qRT-PCR, ELISA and NTA to simply characterization of the isolated sample. I am not sure whether the isolated sample is EV or not. I think it is necessary to make full detailed analyses of the isolated sample step by step.

We have conducted the additional EV characterization for sections 2.4 and 2.5, as the reviewer requested. We first highlighted our earlier qRT-PCR, ELISA and NTA characterization data by specifying the manuscript and SI figures etc in sections 2.3 and 2.4. We had also conducted extensive cholesterol quantification of HDL and LDL to demonstrate the purity of the isolated EVs. Sentences are added to section 2.4 to emphasize NTA characterization of the EV before and after lipoprotein isolation shows high yield and high-quality EVs without lysis or coalescence. Another sentence was added to point out additional benchmarking against a commercial magnetic bead immunocapture technology. The new WB characterization is now added to Figure 5 and discussed in 2.5 to demonstrate the isolated sample is EV. We hence believe we have conducted extensive quantitative and qualitative characterization that shows the isolated sample is EV and isolation from HDL/LDL is done with high yield and quality.

In Figure4, Cholesterol was decreased after MNM purification (about loss 6/7 after MNM purification), but the number of EVs did not significantly decrease. Why does the plasma have high level of LDL cholesterol?

Plasma LDL does outnumber EVs by 1000 fold (microM vs nM). That is exactly the main purpose of our MNM EV isolation technology, to purify specific EVs from the more abundant lipoproteins like HDL, LDL and VLDL. Of course, a secondary result is that the lipoproteins are also isolated and purified from the EVs. EV and lipoprotein nanocarriers may carry different protein and nucleic acid biomarkers, with different clinical implications. The lipoproteins are captured by the nanobeads and removed from the samples through MNM, thus the cholesterol amount decreased in the purified EV sample in the MNM flow through, as EVs do not carry cholesterol. We note that HDLs are small and not visible in NTA and hence specific immune nanobeads are used capture both LDL and HDL. The characterization is to demonstrate that ANM can remove

lipoproteins from the samples without interfering with the purified EVs. We have added sentences in the sections 2.3 and 2.4 to clarify these issues.

Figure 5, EGFR protein is notably decreased after MNM purification compared with ANM isolate. Why?

In the inlet of Fig 5b, we compared EVs isolated by the size-based ANM technology and MNM flowthrough of MNM, which is based on antiEGFR immunocapture. Since the ANM EV isolate is passed through the MNM, our results show that we have pulled down specific EV with EGFR from the ANM isolate. This is the main point of section 2.5 and Figure 5. We have now clarified this issue in section 2.5 and caption of Figure 5 with additional sentences.

Minor:

1. What are the HDL samples? I can't find the information.

Collection and isolation of the HDL and LDL samples are now described in 4.7.

Responses to Reviewer #2 :

This paper demonstrated the utility and efficiency of the electroplated magnetic nanoporous membrane (MNM) with unique heterogeneous superparamagnetic junctions and this method could achieve high-efficiency capture of superparamagnetic nanobeads and quantify biomarkers carried by EVs and lipoproteins in heterogeneous physiological fluids.

The theme is of significance and helps researchers in biomedical fields to further enhance the capture efficiency of specific extracellular vesicles and lipoproteins. The content of this article is comprehensive and the skeleton is clear. Thus, I recommend the acceptance of this manuscript in Communications Biology after the following revisions:

We are glad the reviewer finds our work significant and have implemented his/her suggestions to the best of our ability.

1. The nanobeads used in this study should be introduced briefly to provide more information.

We have added information about the nanobeads in section 4.8.

2. Figures 1c and 1d showed the cross-section SEM images of a single nanopore on a sputtered magnetic nanoporous membrane and a single nanopore on the electroplated magnetic nanoporous membrane. It is recommended to reduce the magnification to get an overview of SEM images. The structure of the nanopore edge could be exhibited better if characterized by elemental analysis than highlighted by the blue color block.

We added the full image of the pores without pseudo color in the SI. The EDS of our SEM facility is unable to resolve sub-10nm feature such as the tip of our nanopores. We believe our polymer membrane poses problems for EDS preparation for TEM.

3. As Figure 2c shows, the nanobeads were captured inside a single nanopore by the dipole field of the wedge junction. When the sample was transported at high flux for a long time, could the nano-magnetic beads block the nanopores, and thereby reduce the capture efficiency and the yield of specific isolation?

This is true. The nanobeads will not get trapped once the membrane is saturated, which is around 6mL of 10× diluted nanobeads for our membrane size (varies in different nanobead batches). In all of our biological experiments, we make sure there are no nanobeads flowing through the MNMs. We have added a sentence in section 2.2 to clarify this.

4. The reviewer is wondering whether the extracellular vesicles and lipoproteins captured by MNM can be re-released for subsequent applications.

Developing high efficiency releasing protocol is our next step. We added some observations in this direction at the end of the Discussion section.

5. To make the comparison of the EVs size distribution before and after purification more intuitively, it is suggested to combine Figures 4c and 4d.

We have updated Figure 4 accordingly.

6. There are a lot of minor problems in the figures and should be corrected carefully. Figure 1c is too blurry and some extra borders in other figures should be removed, such as Figures 1g, 1h, 3, 4, and 5.

We have improved the Figures accordingly.

7. The signs of statistical significance in Figures 2e, 3c, and 5b should be well-positioned. In addition, the scale bar in Figure 2d should be clearer and the picture of the actual experimental setup in Figure S2b could be optimized.

We have updated the Figures as instructed. The symbols in some figures are individual data points.

8. Recently, there has been quite a lot of research published on this topic area, especially organ-on-chips. The authors are suggested to read them and use them as references. The following articles may be mentioned:

Science Bulletin 2019, 64, 1110-1117.

Research 2019, 2019, 6906275.

Engineered Regeneration, 2020, 1, 35-50.

Research 2021, 2021, 9829068.

Engineered Regeneration, 2021, 2, 182-194.

Advanced Science 2022, 9, 2104272.

We have read these publications and have added a few at the end of the Discussion section. We agree that biomarker discovery with organ-on-chips or organoids would be another important application of this MNM technology.

Responses to Reviewer #3 :

Zhang et al. designed the utility and efficiency of electroplated magnetic nanoporous membrane with heterogeneous superparamagnetic junctions for immunocapture of HDL particles and specific extracellular vesicles. Generally, the manuscript should be revised in order to make it more accessible to the general audience.

We have implemented the revisions suggested to make the manuscript more accessible.

1. Some figures are not described in the results section of the manuscript, such as Figures 1e and f, Figures 2d.

We have added discussion of these figures in section 2.1 and 2.2. The key concept is that sharp corners can amplify the magnetic field gradient to pull down superparamagnetic beads. Just high magnetic field is not sufficient. The field gradient should also be high.

2. The supporting information in the results section should be marked with the corresponding figure number and explained in detail.

We have marked the corresponding figure number in Supporting Information and added more explanation.

3. The Discussion should be expanded.

We have expanded the Discussion to address future directions and possible applications.

4. In the figure legend of Figure 2e (Line 516), 10 mL/mL should be corrected to 10 mL/hr. In addition, groups should be marked more clearly in this image, which might easily confuse the reader.

We have updated Figure 2 accordingly.

5. In the Figures 4 and 5, it is not enough to use NTA alone to confirm whether the isolated particles are EVs. Transmission electron microscopy images should be supplemented to confirm the morphology of EVs isolated by MNM, and western blot should be used to verify the marker proteins of EVs. In addition, since EGFR is a transmembrane protein, Western blot should be used to verify whether high purity EGFR-specific EVs can be isolated by the MNM.

We have now added western blot experiment to validate our EGFR-specific EV capture in section 2.5 and in Figure 5. We currently lyse the pulled-down EV samples directly on the MNM membrane and characterize the contents. TEM imaging of the pulled-down EVs cannot be done on our membrane. Developing high efficiency releasing protocol is our next step, as we now emphasize at the end of the Discussion section, and TEM imaging will be conducted then.

Reviewers' comments:

Reviewer #1 (Remarks to the Author):

The manuscript is improved. Only one concern:

The characterization of the isolated EVs is still poor in 2.4 and 2.5. At least some very important experiments should be done, such as the expression of EV markers CD63, CD9. And density gradient centrifugation should be used to confirm the sample contains EV.

Reviewer #2 (Remarks to the Author):

I have no further questions now. The manuscript could be accepted for publication.

Reviewer #3 (Remarks to the Author):

In this new version, the authors have made major revisions and addressed all the concerns and questions of reviewers. Therefore, the manuscript should be considered for publication in Communications Biology. However, in order to further improve the quality of the manuscript, the figures in the manuscript still need to be further optimized. The fonts on many figures are not clear, such as Figure 3c and 3d. Please try to improve the resolution of the figures or increase the size of the font.

Reviewer #1 (Remarks to the Author):

The manuscript is improved. Only one concern:

The characterization of the isolated EVs is still poor in 2.4 and 2.5. At least some very important experiments should be done, such as the expression of EV markers CD63, CD9. And density gradient centrifugation should be used to confirm the sample contains EV.

We are glad that the reviewer found the revised manuscript improved and had only one remaining concern that the isolated nanocarriers are EVs. In addition to our earlier NTA verification that the captured nanocarriers are in the size range of EVs and the elimination of lipoproteins as captured nanocarriers (Figs 3 and 4), we have now conducted additional ELISA quantification of CD63 and CD9 EV markers for both the lipoprotein pulldown and the EGFR EV pulldown, as the reviewer suggested. These new data are now reported in the revised sections 2.4 and 2.5, and in the new ELISA data in the inset of Figure 4c and in Figure 5c. In addition, we report in Fig. 5e Western Blot of syntenin 1 in the EGFR isolate, another EV marker. As discussed in the revised sections, the CD63 and CD9 ELISA, the miR21 qRT-PCR and the NTA size spectrum all produced quantitatively consistent results before and after MNM treatment. These abundant data verified that the isolated nanocarriers are EVs. All of the samples are purified by our asymmetric nanoporous membrane (ANM, ref [33]), which surpass centrifugation-based method in both yield and purity. Another paper on ANM, with detailed characterization and benchmarking, will be ready soon. We hence prefer not to do density-gradient purification, as our previous and new characterization studies have validated that the isolated nanocarriers are EVs.

Reviewer #2 (Remarks to the Author):

I have no further questions now. The manuscript could be accepted for publication.

Reviewer #3 (Remarks to the Author):

In this new version, the authors have made major revisions and addressed all the concerns and questions of reviewers. Therefore, the manuscript should be considered for publication in Communications Biology. However, in order to further improve the quality of the manuscript, the figures in the manuscript still need to be further optimized. The fonts on many figures are not clear, such as Figure 3c and 3d. Please try to improve the resolution of the figures or increase the size of the font.

We are glad that the reviewer 3 confirms that we have addressed the concerns and questions of all reviewers, and that the manuscript is ready for publication in Comm Bio. As he/she suggested, we have improved the fonts in Figures 3c and 3d and have generally improved the quality of the figures.

REVIEWERS' COMMENTS:

Reviewer #1 (Remarks to the Author):

The authors addressed all my concerns. I think it is now can be published.